# Constant light and high fat diet alter daily patterns of activity, feed intake and fecal corticosterone levels in pregnant and lactating female ICR mice

**Leriana Garcia Reis** [ID]**, Kelsey Teeple, Jenna Lynn Schoonmaker** [ID]**, Corrin Davis, Sara Scinto, Allan Schinckel, Theresa Casey** [ID] *

Department of Animal Science, Purdue University, West Lafayette, IN, United States of America

* theresa-casey@purdue.edu

**Data Availability Statement:** All relevant data are within the manuscript and its Supporting Information files.

## Abstract

The prevalence of constant light exposure and high-fat diet in modern society raises concerns regarding their impact on maternal and offspring health outcomes. In rodents, exposure to maternal high-fat diet or continuous light negatively program metabolic and stress response outcomes of offspring. A 2x3 factorial study was conducted to investigate the impact of diet (control–CON, 10% fat, or high fat–HF, 60% fat) and exposure to different lighting conditions: regular 12-hour light-dark cycles (LD), continuous dim light (L5), or continuous bright light (L100) on female ICR mice daily patterns of time in and out of the nest, feed intake, and fecal corticosterone levels during gestation and lactation. Our previous analysis of these mice found HF diet decreased number of pups born, but increased litter growth rate to postnatal (PN) d12. Whereas continuous light increased gestation length and tended to increase PN litter growth. Here we report that patterns of grams of feed intake, an indicator of feeding activity, were affected by light, diet, period of the day (day versus night) and physiological state (gestation and lactation), with significant interactions among all these variables (P<0.05). HF diet and light treatment increased fecal corticosterone output (P<0.05) during lactation. Dams exhibited significant 12 h and 24 h rhythms of activity out of the nest in the first 48 h postnatal, with time outside of the nest greater in the second 24 h period. L100 treatment and HF diet attenuated rhythms and shifted phase of rhythms relative to LD and CON, respectively (P<0.05). Alterations in behavior affect maternal physiology, including level and timing of release of corticosteroids. Elevated fecal corticosterone levels due to high-fat diet and continuous light may have potential implications on maternal-offspring health, and potentially underlie some of the adverse effects of modern lifestyle factors on maternal and offspring health.

## Introduction

The escalating rates of obesity, among women of childbearing age pose significant health challenges, including decreased fertility and adverse pregnancy outcomes, with prepregnancy

**Funding:** L.G.R. receveid financial support (Grant n˚ 2021/12819-0) as Research Internships Abroad (BEPE) from São Paulo Research Foundation (FAPESP). https://fapesp.br/en The funders had no role in study design, data collection and analysis, decision to publish, or preparation of the manuscript.

**Competing interests:** The authors have declared that no competing interests exist.

obesity increasing the risk for gestational diabetes, hypertension, and preterm delivery, along with heightened neonatal morbidity and mortality [1]. Also potentially increasing the risk for adverse maternal health outcomes are modern lifestyle factors like shift work and continuous exposure to light at night [2]. Continuous exposure to light, particularly during overnight shifts, disrupts circadian clocks. Circadian clocks regulate nearly all physiological systems, generating 24-hour rhythms of hormones, metabolism, and behavior, functioning as a critical regulator of homeostasis by synchronizing internal physiology to the external environment [3]. In mammals, circadian clocks are regulated hierarchically, with the master clock located in the suprachiasmatic nuclei (SCN) of the hypothalamus. The light-dark cycle is the primary environmental cue that entrains the timing of SCN, while other inputs include behavior, nutritional status, and stress. The SCN integrates temporal information, including ambient light, and communicates this information to peripheral clocks partly through the regulation of circulating hormones, such as cortisol, to coordinate the timing of physiology across the body. Levels of cortisol begin to rise prior to waking, peak shortly after, then drop across the day. Cortisol is also released in response to stress, and its levels and rhythms are affected by the metabolic and reproductive state of the animal [4–6].

Maternal metabolic adaptations during pregnancy and lactation shape the long-term health of offspring. Experiments conducted using rodent models demonstrate that exposure of offspring to maternal prepregnancy obesity, circadian clock disruption, and stress adversely program metabolic outcomes in later life [7,8]. For instance, rat pups exposed to circadian disrupting conditions during these critical periods develop metabolic syndrome in adulthood [9]. Epidemiological and controlled animal studies report that early exposure to maternal high-fat diet during pregnancy and lactation has long-term effects on offspring health, which include development of metabolic syndrome, hypertension, type 2 diabetes, obesity, cardiovascular dysfunction, and alterations in neurological development [10–13]. Given the prevalence of high-fat diets and nocturnal light exposure in contemporary society, understanding their individual and combined effects on behavior, feed intake, and stress hormone levels is crucial [14–18].

Our previous studies of the female ICR mice used in experiments described herein, found that 4 weeks of high-fat (HF) diet feeding during prepregnancy resulted in greater body weight, body mass index (BMI), and percentage of body fat in HF fed mice compared to control (CON) fed mice [17]. Relative to mice on CON diet, mice on HF diet also increased food consumption during the day, typically a period of lower activity, and had elevated hair corticosterone levels, and disrupted fecal corticosterone circadian rhythms, with higher basal levels, weakened rhythms, and a shift in phase compared to CON mice. These results indicate that high-fat diet-induced obesity, disrupted normal feeding patterns, and disturbed circadian rhythms of corticosterone in female ICR mice, highlighting potential adverse effects on circadian and metabolic regulation. Following these mice after being bred to males demonstrated that HF diet reduced number born while increasing postnatal day 12 litter weight compared to the CON diet [19]. During pregnancy, HF fed mice consumed fewer grams of food per day but had higher average daily caloric intake compared to CON mice, which was related to greater rates of litter growth to peak lactation. In this 2x3 factor designed study, mice were also assigned to one of three light exposure treatments: control 12h light to 12h dark cycle (LD), continuous low lux (L5), or high lux (L100) light. Continuous light exposure prolonged gestation, affected dam feed intake, increased serum prolactin, final dam and mammary gland weight, while decreasing mammary ATP content and milk lactose. Exposure to continuous light also tended to increase litter weight regardless of diet. Correlation analysis revealed positive relationships between final litter weight and mammary size, metabolic stores, kilocalories (kcal) of feed intake, and gestation length. Overall, the HF diet impaired reproductive

outcomes, such as litter size, and influenced feeding behavior and metabolism. Although some of the effects of constant light exposure appear to be associated with circadian clock disruption, such as prolonged gestation and lower milk lactose levels, we proposed that a potential long day photoperiod response also occurred, which was indicated by higher circulating levels of prolactin and increased body and mammary weight of females exposed to these conditions [15].

The objective of this study was to determine the effect of HF diet and continuous light exposure on daily patterns of eating throughout gestation and lactation, activity outside of the nest during early lactation, and fecal corticosterone output during lactation to gain a better understanding of how changes in nutritional status and ambient light affect maternal behavior and corticosterone levels, as both are crucial for offspring development. While we recognize that the diets differ significantly in energy concentration (kcal), our focus is on understanding how these changes influence behavior, specifically the amount of food consumed and the timing of eating, and how these factors affect the physiology of behavior. We hypothesize that a high-fat diet combined with continuous light exposure will result in altered behavioral patterns, increased corticosterone levels in feces, and modified feed intake compared to the control diet.

## Material and methods

### Experimental design

The animal experiments were reviewed and approved by Purdue University's Institutional Animal Care and Use Committee (protocol 2104002135) prior to beginning experiment. The experiments described herein were part of a larger study described by us previously [15,17]. Female ICR mice (n = 87) were obtained from Envigo (Indianapolis, IN) at three weeks of age, underwent a 2-week acclimation period, and then were randomly assigned to control (CON, n = 36) or high-fat (HF, n = 49) diets. HF group had more females due to expected decreased fertility [20]. Initial weights were similar (CON = 21.7 g ± 0.52, HF = 21.7 g ± 0.45), and mice were group-housed (3–5 per cage) with *ad libitum* access to food for 4 weeks, inducing body mass and fat divergence pre-mating [17]. Diets were matched for sucrose content (7%) but differed in fat (CON: 10% energy from fat, HF: 60% energy from fat) and carbohydrate composition. Fat sources included soybean oil, which was primarily in CON, and lard was mostly HF, with varying proportions between diets. Detailed diet compositions are available from Research Diets, Inc., and previously published work [21]. Following dietary manipulation, females were paired with males and monitored for presence of vaginal plugs twice daily. Upon vaginal plug observation or after 5 days with the male, females were assigned to one of two light conditions: 12 hours light:12 hours dark (LD), continuous dim light (L5), or continuous bright light (L100). Light intensity was measured (LD: 114±13.78 lux, L5: 3±1.51 lux, L100: 106.25±7.91 lux). Mice remained under specified conditions until the study's end on day 12 of lactation (Fig 1). This timing coincided with pups' developmental milestones, opening their eyes and consuming maternal food. Pregnancy day 1 was determined by vaginal plug detection.

### Feed intake

Dam feed intake was measured daily throughout the study, which ended on lactation d12. Feed intake was assessed by measuring consumption during the day between 0530–1730, which was the light phase for the LD group, and during the night between 1730–0530, which corresponded to the dark phase of the LD group. Individual mouse feed weights were recorded Monday through Friday at 0530 and 1730. Pregnancy feed intake was analyzed as first third of gestation (d1-6), second third (d7-13), and last third (d14-18) and lactation feed intake was analyzed across three periods: days 1–4 (early), 5–8 (middle), and 9–12 (peak). Energy intake

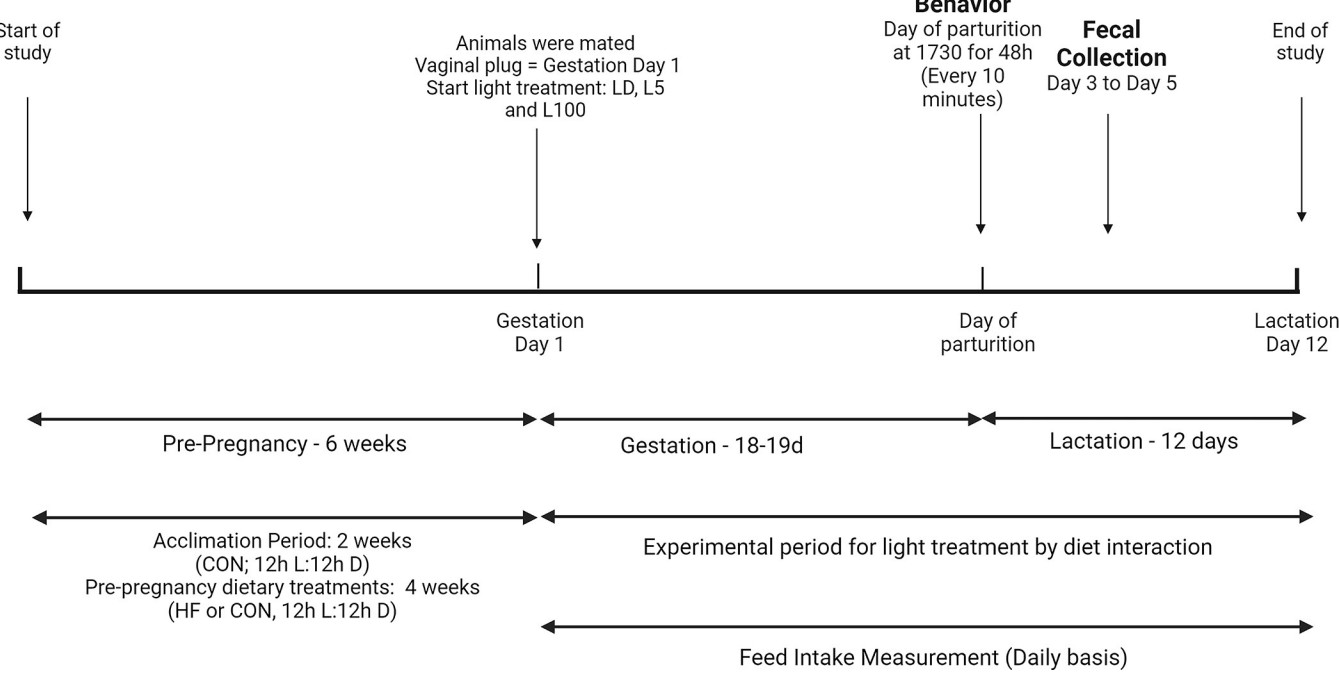

**Fig 1. Experimental timeline overview.**

(kcal) was calculated by multiplying consumed grams by 3.85 kcal/g for CON and 5.24 kcal/g for HF.

## Fecal collection, corticosterone extraction, and analysis using enzyme-linked immunosorbent assay (ELISA)

Fecal material was collected twice daily from day 3 to day 5 over lactation to capture day (0530–1730) and night (1730–0530) output. Mice were transferred to a new cage with fresh corn cob bedding, while the old bedding was retained. Fecal material was separated from the bedding, collected, and stored at -20˚C. The weight of fecal material was measured to determine the quantity produced over the day and night period. Corticosterone was extracted per Arbor Assay protocol. Approximately 0.2 g of fecal material was weighed and then crushed into a fine powder using Precellys 24 Lysis & Homogenization (Bertin Technologies, Montigny-le-Bretonneux, France). After grinding, 2 mL of ethanol was added and the sample was centrifuged. Approximately 1.3 mL of the supernatant was pipetted into 2 separate tubes and dried extracts were stored at -20˚C. Samples were reconstituted in absolute ethanol and analyzed using the corticosterone Enzyme Linked Immunosorbent Assay (ELISA) from Arbor Assays (catalog no. K014, Ann Arbor, MI, USA) following manufacturer's protocol. Absorbance was read at 450 nm on the Spark 10M multimode microplate reader (Tecan Trading AG, Switzerland).

## Behavior over the lactation

A subset of mice (CON-LD, n = 4; HF-LD, n = 3; CON-L100, n = 3; HF-L100, n = 3) were continuously video recorded using infrared closed-circuit television cameras (CCTV) (Sony,

Tokyo, Japan) and GeoVision software (Taipei, Japan). The subset of mice selected were random within treatment groups and based on number of available cameras. Following capture, research assistants reviewed video to identify day of parturition, and began marker behavior beginning at 1730 on day of parturition and continued for 48 hours. A binary coding system was employed, where a score of 0 was assigned when the dam was observed inside of the nest, indicative of maternal behavior or rest, and a score of 1 was attributed when the dam was observed out of the nest, indicating active behavior outside of maternal care or rest. Behavior was data collected every 10 min by forwarding through video. The percentage of time out of the nest was calculated by dividing the total counts in an hour by 6 and multiplying by 100.

## Statistical analyses

All statistical analyses were performed using SAS 9.4 (SAS Inst. Inc., Cary, NC), except for analysis of feed intake data, which was performed using R version 4.1.2 [22]. The data were analyzed in a completely randomized design, and each mice was considered an experimental unit. The normality of the residues was verified by the Shapiro-Wilk test (UNIVARIATE procedure of SAS), and the Levene test compared the homogeneity of the variances. Residuals for behavior data did not follow a Gaussian distribution, so a generalized linear model was used with Poisson distribution. Variables with a continuous distribution such as feed intake and corticosterone concentration were analyzed using the MIXED procedure of SAS. Statistical model included treatments (diet and light), period of the day (day or night) as fixed effects, and mice as random effects. The SLICE option using the LSMEANS/PDIFF command was used to explore the interactions of data collection. The Kenward-Roger method was used to correct the degree of freedom of the denominator for the F test. The covariance structure was determined based on the lowest Akaike (AIC) information criteria value.

Regression analysis was performed to evaluate out of the nest behavior, using the treatment effects (diet and light) and period of the day, continuous variables X1, X2, X3, X4, X5, and X6, which were incorporated into a Fourier series model. The Fourier series model variables X1 and X2 represented the sine and cosine single phases (24 h), X3 and X4 represented the sine and cosine double phases (12 h), and X4 and X6 represented the sine and cosine triple phases (8 h) of shifts in the curve [23,24]. The regression analyses were completed using the PROC MIXED procedure including the main effects of treatment (diet and light), day, and periodic regression variables, whereas mice and hour as a random effect. The regression analysis examined the interactions of treatments with the sine and cosine functions. Significance level was set at $P < 0.05$ and tendency towards significance at $0.05 < P < 0.10$.

## Results

We previously reported that diet ($P < 0.001$) and stage of reproduction (i.e. pregnancy, parturition, and lactation) significantly ($P < 0.001$) affected daily feed intake, measured in grams and kcal consumed per day [15]. The level of intake was the lowest at the beginning of pregnancy, increased until the end of lactation, and briefly dropped on the day of parturition. CON mice consumed more grams ($3.92 \pm 0.10$ g and $9.24 \pm 0.18$ g) than HF ($3.36 \pm 0.10$ g and $8.27 \pm 0.17$ g) during pregnancy and lactation, respectively. However, HF fed mice had greater daily kcal intake ($17.6 \pm 0.47$ kcal and $43.4 \pm 0.76$ kcal) than CON ($15.1 \pm 0.47$ kcal and $35.6 \pm 0.82$ kcal) during pregnancy and lactation, respectively ($P < 0.05$). A significant interaction between diet and light was also evident during pregnancy for both grams and kcal of diet consumed, with CON-L100 mice consuming a greater number of grams ($4.48$ g $\pm 0.17$) than CON-LD ($3.73 \pm 0.18$ g), CON-L5 ($3.54 \pm 0.17$ g), HF-LD ($3.10 \pm 0.20$ g), HF-L5 ($3.64 \pm 0.16$ g), and HF-L100 ($3.32 \pm 0.17$ g). CON-L100 consumed more kcal ($17.2 \pm 0.81$ kcal) than

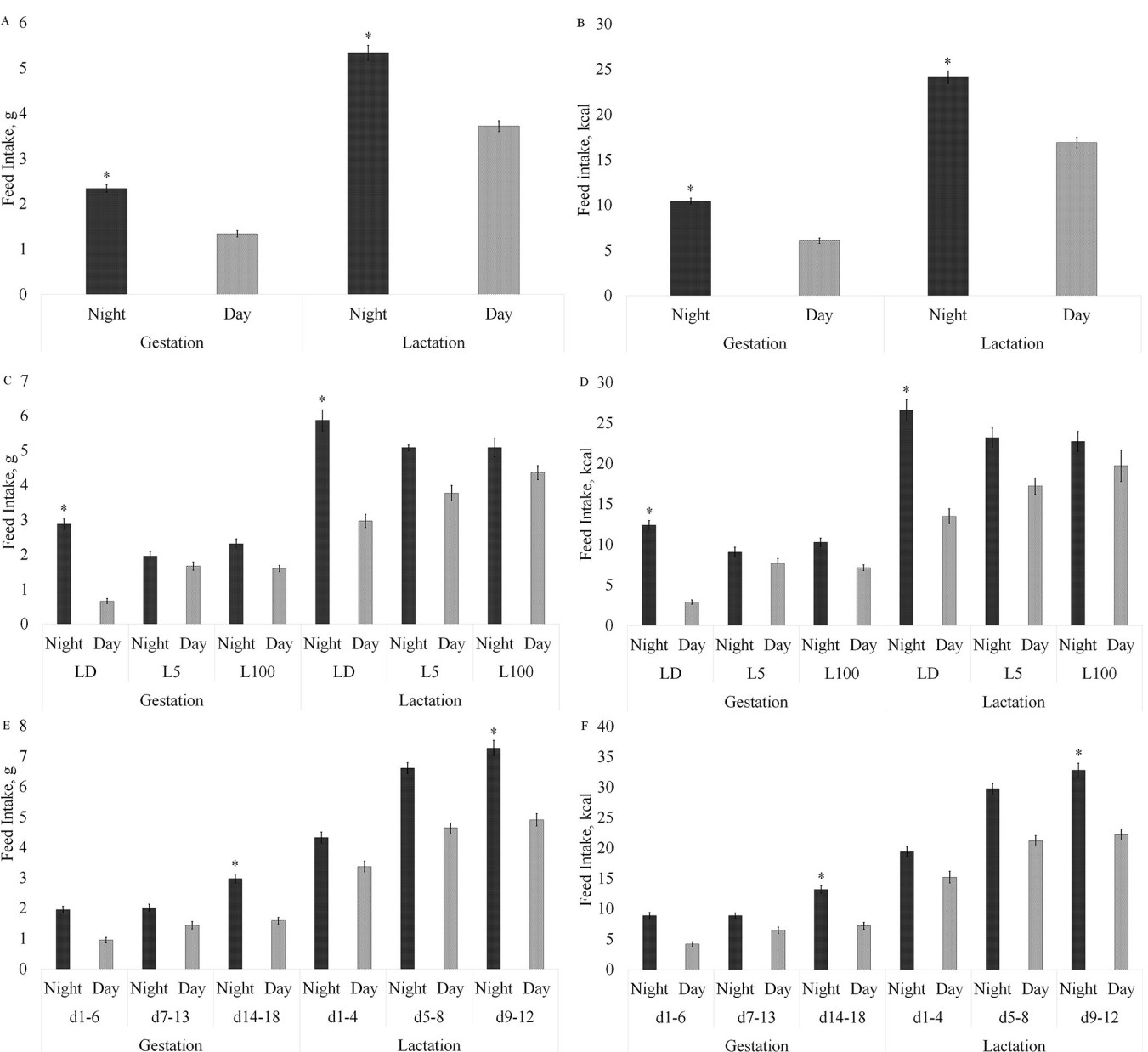

**Fig 2. Effect of diet, light treatment, and physiological state on the period of day of feed intake in mice across gestation and lactation phases, differentiating day and night periods.** Data represent mean ± standard error of the mean (SEM). Statistical analysis indicates as significant (P<0.05) period of the day, light*period of the day, term*period of the day, for both grams and kcal, considering gestation and lactation phase. *Significant effect showing the period of the day or the interaction of light or physiological state with the period of the day where there was the highest feed intake.

CON-L5 (13.6 ± 0.79 kcal). CON-L5 mice also consumed less kcal than HF-L5 (19.1 ± 0.73 kcal) and HF-L100 (17.3 ± 0.78 kcal). Lastly, CON-LD mice consumed less (14.4 ± 0.84 kcal) than HF-L5 mice (19.1± 0.73 kcal). The significant diet and light interaction did not continue into lactation.

Herein we report on the effect of diet, light treatment, and physiological state on the period of day of feed intake to understand how these variables affect maternal feeding behavior (Fig 2). Period of the day had an overall influence on food consumption (grams and kcal) (P<0.001), with the highest intake occurring at night (Fig 2A and 2B). There was no

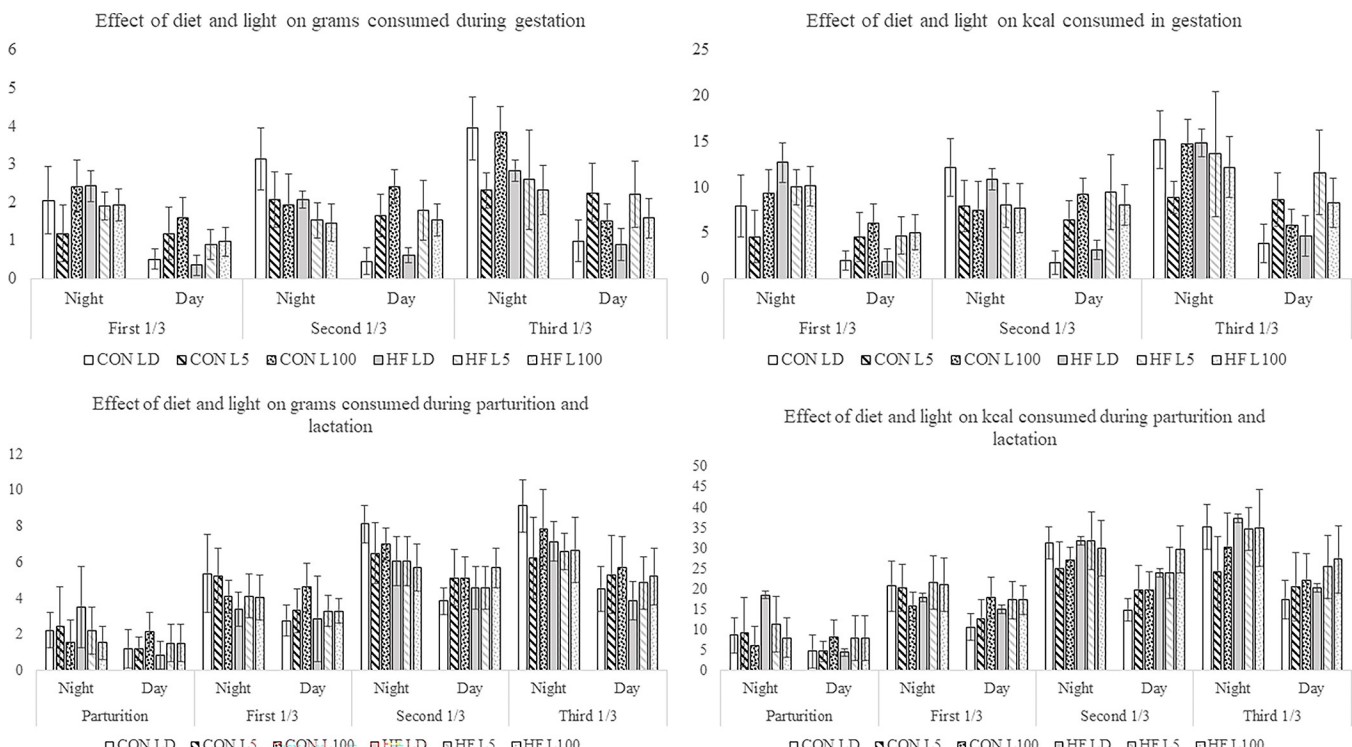

**Fig 3. Interaction effects of diet, light, physiological state, and time of day on feed intake during pregnancy and lactation.** Data represent mean ± standard error of the mean (SEM). Effect of diet and light on feed intake (grams and kcal) during pregnancy and lactation. Statistical analysis indicates many significant (P<0.05) interactions among diet, light, period of the day, and stage of reproduction (S1 File).

interaction between diet and period of day (P>0.05). However, there was an interaction between light and the time of day revealed. The LD group exhibited greater feed intake at night than during the day during both gestation and lactation, in terms of both grams and kcal (P<0.001) (Fig 2C and 2D). The LD group drove the overall difference in intake between the light and dark period. Furthermore, feed intake was influenced by an interaction between physiological state and period of day, with mice consuming more feed (grams and kcal) at night during gestation days 14–18 and lactation days 9–12 (Fig 2E and 2F).

A complex interaction was evident between diet, light, time of day, and physiological state on feed intake (Fig 3), with 189 significant (P<0.05) interactions for diurnal feed intake (grams) during gestation and 464 significant (P<0.05) interactions in the lactation stage of the experiment (S1 File). Notably, the CON L5 group exhibited no differences in feed consumption (grams or kcal) between day and night during any period of pregnancy or lactation, whereas the CON L100 group exhibited changes in the patterns of feed intake between day and night periods during gestation. In early gestation CON L100 consumed more (grams and kcal) during the night, in the second third of gestation they consumed more during the day, and in the last third of gestation CON L100 mice consumed more during the night. However, during lactation, CON L100 consumed the same amount of feed in the day and night across all 3 periods of lactation. During the first third of pregnancy, HF L5 and HF L100 consumed more (in grams and kcal) at night. However, from the second third of pregnancy to the end of lactation, HF L5 and HF L100 consumed the same amount in the day and night periods.

The interaction between diet, light, time of day, and physiological state significantly impacted feed intake in kcal. Specifically, the CON L100 group displayed a noticeable shift in

feed consumption patterns: higher intake during the night in the first and last thirds of gestation and higher intake during the day in the second third (Fig 3). In the HF L5 and HF L100 groups, feed consumption in kcal equalized between day and night from the second period of pregnancy onwards. During lactation, the CON L100, HF L5, and HF L100 groups all showed consistent feed intake in kcal between day and night (Fig 3).

Diet and period of the day had an overall effect on fecal output (g), where mice on HF diet (0.72 ± 0.03 g) produced more feces than CON (0.59 ± 0.03 g), and more feces was produced in the night period (0.70 ± 0.03 g) of the 24h cycle, than the day (0.61 ± 0.03 g, Table 1). A significant interaction between light treatment and period of the day was found for fecal output (P = 0.003, Fig 4). There was no difference in fecal output between the day and night in L5 and L100 treatments, whereas LD mice defecated more during the night than day. The concentration of corticosterone (pg.mL$^{-1}$) in feces was not affected by diet, light treatment, period of day (day or night), or any treatment or period of the day by treatment interaction (P>0.05, Table 1). Corticosterone output (pg.g$^{-1}$) represents the total corticosterone measured in the collected fecal samples in a 24h period. HF mice released approximately 27% more corticosterone in their feces (7239.66 ± 403.34 pg.g$^{-1}$) compared to CON mice (5692.83 ± 482.92 pg.g$^{-1}$). Light treatment also exerted a significant influence on corticosterone output, with more released in feces in 24h in the L5 and L100 groups compared to LD (P = 0.029, Table 1). A diurnal variation in total fecal corticosterone output was also evident, with corticosterone output approximately 26% higher during the night compared to the day (P = 0.02).

Diet, light, and period of the day did not affect the percent of time mice spent out of the nest across the 48h period activity was measured in lactating animals (Table 2). However, a significant interaction between light exposure and time period of the day was observed (P = 0.003), revealing that lactating dams exposed to LD spent more time out of the nest in the night period than L100 mice (Fig 5). There was also a significant (P<0.017) increase in the percent of time out of the nest in the last 24 hours of observation (day 2) relative to the first 24 hours (day 1).

**Table 1. Concentrations of corticosterone in feces and fecal output (weight) of female mice during day 3–5 of lactation.**

| | Diet[a] (± SEM[d]) | | Light[b] (± SEM[d]) | | | Period of day[c] (± SEM[d]) | |
| --- | --- | --- | --- | --- | --- | --- | --- |
| | CON | HF | LD | L5 | L100 | Day | Night |
| Corticosterone, pg.mL$^{-1}$ | 2,088.00 ± 150.01 | 2,042.36 ± 125.95 | 1,961.04 ± 168.46 | 2,016.96 ± 182.72 | 2,217.54 ± 156.72 | 1,933.05 ± 138.50 | 2,197.31 ± 138.50 |
| Feces weight, g | 0.59 ± 0.03 | 0.72 ± 0.03 | 0.62 ± 0.04 | 0.63 ± 0.04 | 0.72 ± 0.03 | 0.61 ± 0.03 | 0.70 ± 0.03 |
| Corticosterone output, pg·g$^{-1f}$ | 5,692.83 ± 482.92 | 7,239.66 ± 403.34 | 5,590.09 ± 549.04 | 6,219.29 ± 578.74 | 7,589.34 ± 504.37 | 5,705.70 ± 438.70 | 7,226.79 ± 451.05 |
| | *P-value[e]* | | | | | | |
| | Diet | Light | Period of day | Diet*Light | Diet*Period of day | Light*Period of day | Diet*Light*Period of day |
| Corticosterone, pg.mL$^{-1}$ | 0.817 | 0.505 | 0.184 | 0.505 | 0.458 | 0.839 | 0.857 |
| Feces weight, g | **0.004** | 0.080 | **0.045** | 0.779 | 0.730 | **0.003** | 0.256 |
| Corticosterone output, pg·g$^{-1}$ | **0.018** | **0.029** | **0.020** | 0.798 | 0.791 | 0.150 | 0.219 |

[a]CON, 10% energy from fat; HF: 60% energy from fat.

[b]LD, 12 hours light:12 hours dark; L100, continuous bright light.

[c]Day, defined as the beginning of the day, lights on at 0530; Night, defined as the beginning of night, lights off at 1730.

[d]SEM: *Standard error of the mean*.

[e]Significant results (P<0.05) are marked in bold.

[f]Corticosterone output: (Feces weight x Corticosterone from ELISA) / Feces weight used in extraction).

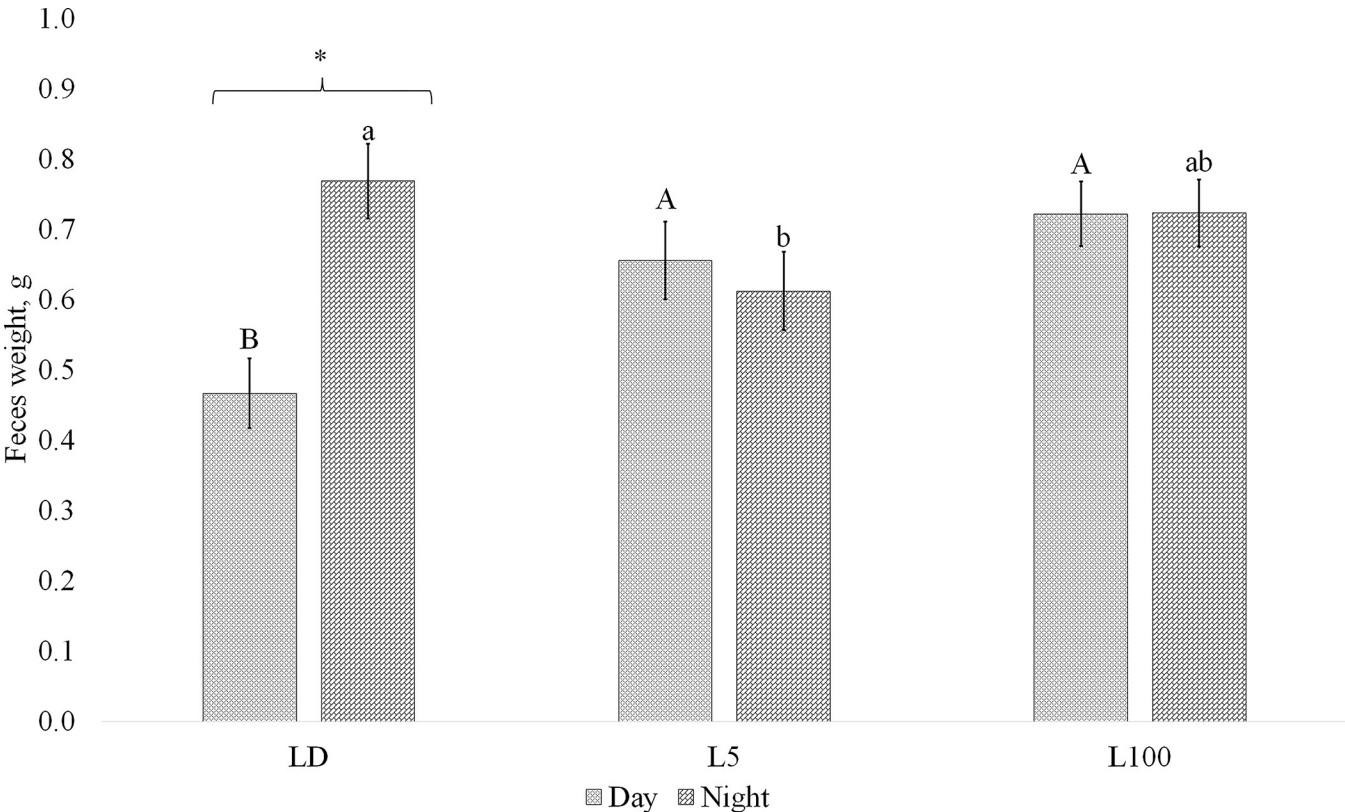

**Fig 4. Interaction between light treatment and period of the day on fecal output.** Mice were exposed to one-of three lighting conditions: 12 hours light:12 hours dark (LD) or continuous low lux (L5) or high lux (L100) of light, and fecal material was collected twice daily to capture fecal output during the day (0530–1730) and night (1730–0530). Data represent LSmean ± standard error of the mean (SEM) of sampling from day 3 to day 5 of lactation. Different uppercase letters indicate significant difference in fecal output during day time hours between light treatments. Different lowercase letters indicate significant difference in fecal output during night time hours between light treatments. An asterisk above the bar, indicates diffence between day and night within a light treatment.

Regression analysis of Fourier models found significant one phase (24 h) and two phase (12 h) cosine variable fitting of percent time out of the nest during lactation (P<0.05). The X2 variable was significantly different between LD and L100 treated mice, indicating a significant shift in phase of the 24 h rhythm (P<0.001), with the peak of the time out of the nest in L100 animals at 0830 and 1730 and in LD at 2030 (Fig 6A). Diet significantly impacted the X3 variable, indicating 12h rhythms were distinct between HF and CON fed mice (P = 0.04). Overall, during the same period between 1530 and 2330, the average time out of the nest was 24.8% for CON and 18.1% for HF (Fig 6B).

## Discussion

During pregnancy and lactation, offspring survival becomes a priority of female mammals, and changes in their behavior and physiology reflect responses to fetal and neonatal metabolic needs. Our previous studies indicated that during lactation the dam's circadian system responds the increased metabolic demands of milk production, as well as metabolic and behavioral cues initiated by the neonate [25]. Moreover, we found that caloric density of the dam's diet affected the amount of food intake (grams) consumed each day [17]. We also found that prepregnancy HF diet alters timing of feed intake, increasing consumption during daytime-light phase, as well as alters circadian rhythms of fecal output, fecal corticosterone output

**Table 2. Mean percent of time mice were observed outside of the nest during the first 48 hours of lactation.**

| Variables | | Mean ± SEM, % | P-value |
|---|---|---|---|
| Diet | CON | 19.36 ± 1.56 | 0.637 |
| | HF | 18.06 ± 1.61 | |
| Light | LD | 19.72 ± 1.61 | 0.535 |
| | L100 | 17.62 ± 1.53 | |
| Day | Day 1 | 16.22 ± 1.44 | **0.017** |
| | Day 2 | 21.41 ± 1.73 | |
| Period of day | Day | 18.72 ± 1.61 | 0.686 |
| | Night | 18.85 ± 1.58 | |
| Light*Period of day | LD*Day | 15.84 ± 2.06 | **<0.001** |
| | LD*Night | 23.41 ± 2.44 | |
| | L100*Day | 22.40 ± 2.53 | |
| | L100*Night | 13.29 ± 1.74 | |

[a]CON, 10% energy from fat; HF: 60% energy from fat; LD, 12 hours light:12 hours dark; L100, continuous bright light; Day 1, first 24 hours of observation; Day 2, last 24 hours of observation; Day, defined as the beginning of the day, lights on at 0530; Night, defined as the beginning of night, lights off at 1730.

[b]Values represented by mean. SEM: Standard error of the mean.

[c]Significant results (P<0.05) are marked in bold.

rhythms, and total corticosterone released in feces across a 24 h period [17]. The objective of this study was to determine the effect of HF diet and continuous light exposure on daily patterns of eating, activity outside of the nest, and fecal corticosterone output during lactation to gain a better understanding of how changes in nutritional status and ambient light affect these variables. The data presented herein demonstrate that maternal HF diet and exposure to continuous light alter feed intake patterns across pregnancy and lactation, patterns of fecal output, the total amount of corticosterone released in feces across the day, and the daily patterns of behavior of dams in and out of the nest in early lactation.

To support the high energetic demands of late gestation and lactation major changes occur in maternal metabolism and eating behavior. Changes in hormones at the onset of pregnancy modify maternal feeding control centers in the hypothalamus to allow hyperphagia, resulting in fat deposition to provide energy stores in preparation for the high metabolic demands of late pregnancy and lactation [26]. Although mice in LD treatments maintained significantly greater feed intake during the night relative to the day, the increase in eating behavior occurred in both the light phase and dark phase of the LD mice. In particular, grams of feed intake of CON-LD mice increased 2-fold between the first third and last third of gestation, and hyperphagia continued into lactation, with dams increasing light phase/day-time feed intake 1.7-fold between the first third of gestation and last third of lactation phase of the experiment. The change in eating behavior of CON-LD decreased the grams of feed intake differential between night and day from 4.4-times in early gestation to 2-fold by 10–12 d of lactation. The changes in eating behavior patterns of mice in LD exposures likely contribute to the overall attenuation of circadian rhythms of metabolic hormones like glucocorticoids in pregnant and lactating rodents, and potentially circadian core clock genes' expression in the mammary gland, as feeding time appears to be an input to the molecular clock in the mammary gland [25,27–30].

CON mice exposed to continuous dim light, CON-L5, consumed a similar number of calories and grams in the daytime and nighttime hours across all physiological stages. Under continuous high lux of light, CON L100 exhibited differences in feed intake between day and

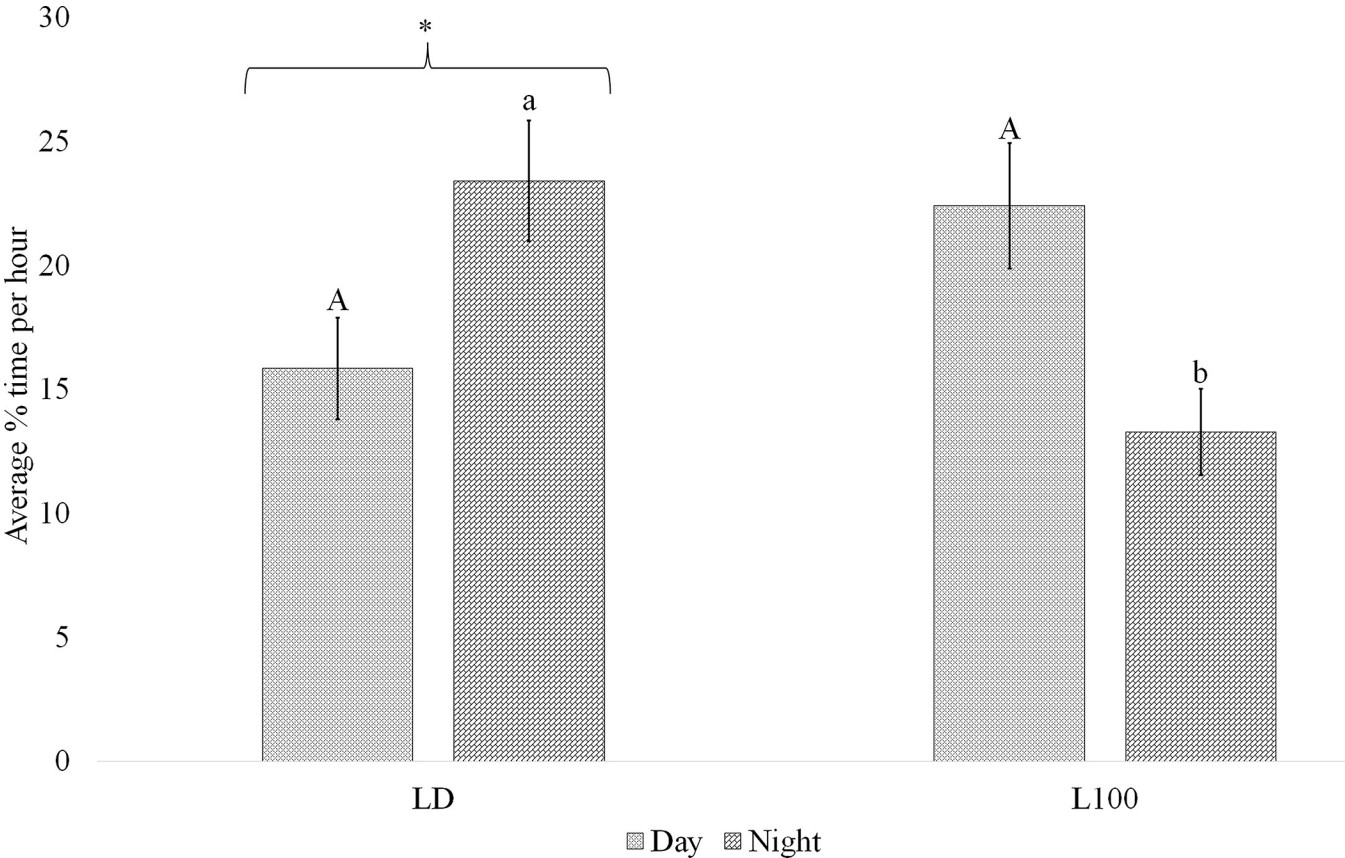

**Fig 5. The interaction between light and period of the day on percent time out of the nest during the first 3 days of lactation.** Mice were exposed to 12 hours light:12 hours dark (LD) or continuous high lux (L100) of light. Time outside the nest was assigned a binary score of 0 (inside nest) or 1 (outside nest) and data were collapsed and expressed as an hourly percent, and then analyzed as percent of time out of the nest during the first 3 d of lactation. Day was defined as 0530–1730, and night as 1730–0530. Data represent mean ± standard error of the mean (SEM). Different uppercase letters indicate signifcant difference in behavior during day time hours between light treatments (P<0.05). Different lowercase letters indicate signifcant difference during night time hours between light treatments (P<0.05). An asterisk above the bar, indicates diffence between day and night within a light treatment.

nighttime hours, but the patterns flip-flopped between gestation phases, being higher in the night period in the first third and last third of gestation, but not different between day and night periods in mid gestation. Except for early gestation, HF L5 and HF L100 groups had equal feed intake between day and night. Thus, for both diets, the L5 light treatment attenuated the circadian feeding cycle, whereas patterns of feed intake appeared to change in the L100 light treatment. Continual changes in behaviors, like varying times of sleeping, appear to be more detrimental to the health of humans [31]. Thus, the negative consequences of exposure to continuous light may be induced by the changing patterns of behaviors of mice, including feeding cycles, and their downstream effectors.

The impact of diet and light exposure on adiposity during lactation is an important consideration, particularly given the energy demands associated with this period. Our prior research reports regarding these female ICR mice presented data that showed HF diet during prepregnancy led to increased body weight, BMI, and body fat percentage, indicating a clear diet-induced obesity phenotype [17]. We observed that despite the prepregnancy adiposity induced by the HF diet, lactating dams experienced changes in body composition due to the demands of lactation. Continuous light exposure (L5 and L100) combined with the HF diet increased body and mammary weight but did not maintain prepregnancy adiposity levels. These

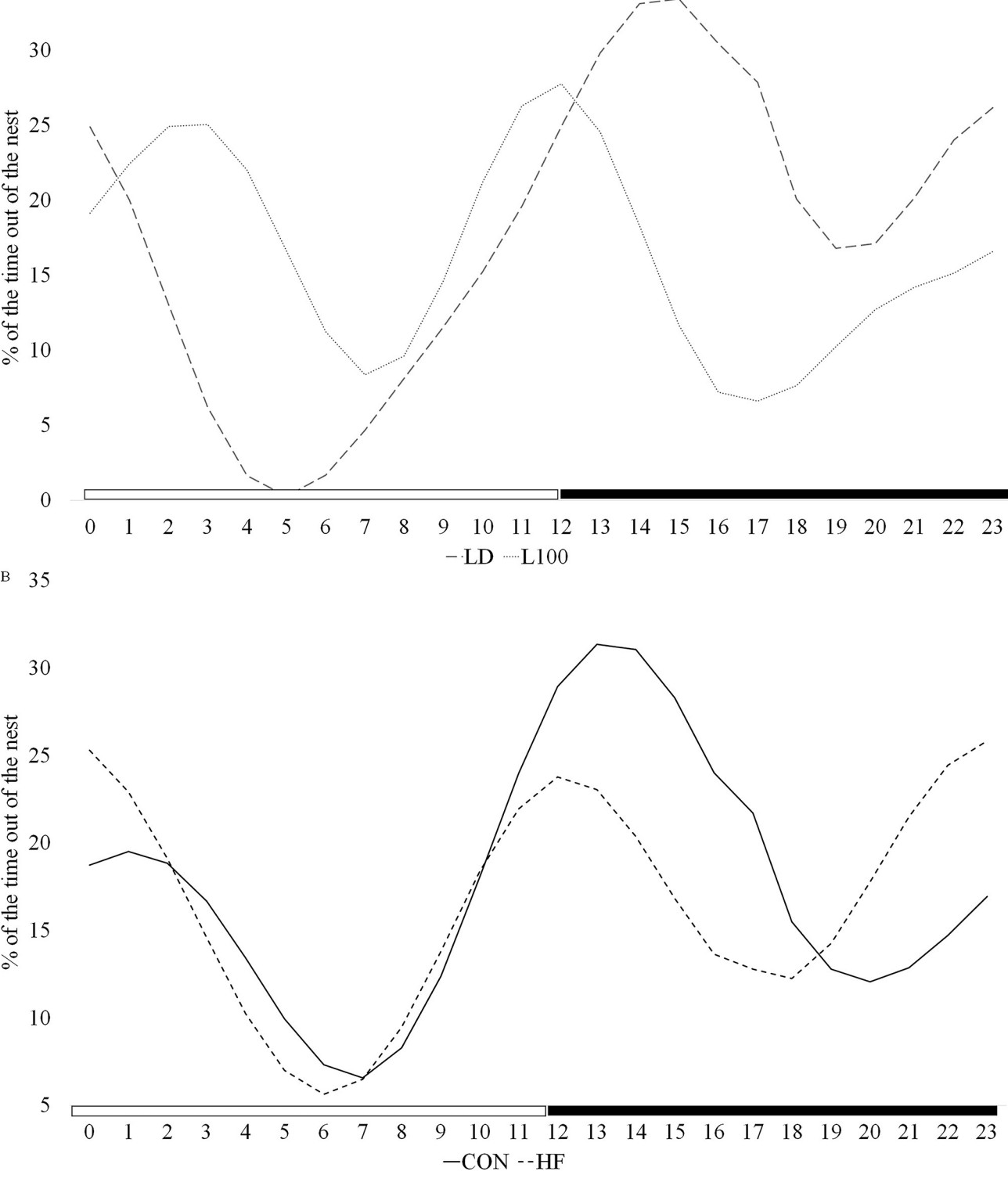

**Fig 6.** Impact of light (A) and diet (B) on percent of time out of the nest during lactation. Mice were exposed to 12 hours light:12 hours dark (LD) or continuous high lux of light (L100) and fed a CON or HF diet of light. Continuous video recordings of the last three days of gestation were reviewed at 10 min intervals, and time outside the nest was assigned a binary score of 0 (inside nest) or 1 (outside nest). Data were collapsed and expressed as an hourly percent, and then Fourier series model variables were calculated and analyzed for fit to cosine and sine curves. Regression analysis indicated Day (P = 0.007), X3*Diet (P = 0.0436), X2*Light (P = <0.001) X2 (P = 0.0375) and X4 (P<0.001) were different.

findings suggest that while the HF diet establishes a higher baseline body fat, lactation and disrupted circadian rhythms due to constant light exposure alter adiposity patterns, influencing reproductive and metabolic outcomes.

Light treatment and diet impacted fecal output patterns and the total daily fecal corticosterone output during lactation. The diurnal differences in feeding patterns of LD treated mice matched differences in fecal output by period of the day, with level of nighttime defecation greater than daytime. In mice, the circadian rhythm of the hypothalamic-pituitary-adrenal (HPA) axis typically results in elevated corticosterone levels during the night [32,33]. This increased nocturnal corticosterone output is linked to higher food intake and activity levels in mice, which facilitates the mobilization of energy stores to meet the heightened metabolic demands [34,35]. Prepregnancy analysis of fecal corticosterone rhythms found mice on CON and HF diets exposed to regular 12 h LD cycles, exhibited circadian rhythm of fecal corticosterone concentration [17]. Here we found no difference between day and night corticosterone concentration, despite differences in fecal output, eating behavior, and time inside and outside the nest by period of the day. Lack of a difference may potentially be explained by differences in sampling protocols between the experiments, physiological state, or both.

In the first experiment, prepregnancy fecal samples were collected every 4 h to capture circadian variation, which was enabled by group housing of 4–5 animals per cage [17]. Herein, mice were housed individually during gestation, and with their litter during lactation when feces were collected. In order to collect enough feces for extraction and analysis within 48 h, a 12 h sampling window was required in this experiment. Collecting fecal samples every 12 h may have obscured any difference in concentration of corticosterone in the day time relative to night time hours of the CON-LD group. The lack of a difference in corticosterone concentration may also potentially reflect the attenuation of rhythms in lactating rodents. During late gestation and early lactation, there is an attenuation in the HPA axis that is believed to function to protect the offspring from potential negative effects of elevated maternal stress hormones during the critical period of early development [36–38].

The higher caloric density of the high-fat diet was related to higher fecal output and greater fecal corticosterone output, with animals on the HF diet showing 27% higher fecal corticosterone output compared to those on the CON diet. In the prepregnancy period animals on the HF diet also exhibited greater daily fecal corticosterone output than those on the CON diet [17]. Levels of hair corticosterone were also greater in animals on HF diets prepregnancy and at the termination of the study on lactation day 12 [15,17]. Together data indicate that HF diet increased basal levels of corticosterone in mice prior to and after the onset of pregnancy and into lactation. Studies of male mice indicate that high fat diet induce a form of chronic stress reflected in elevated corticosterone levels [39], and so elevated fecal corticosterone output likely reflects this stress.

Total daily fecal corticosterone output was also greater in the L100 group relative to other light treatments. This finding is consistent with constant high lux light increasing stress in adult mice, accompanied by elevated plasma corticosterone level [40]. However, others reported 3 weeks of constant light exposure in male mice depressed plasma corticosterone levels [41]. Analysis of the impact of constant light on maternal plasma corticosterone levels found no differences between this group and the control light-dark group. The discrepancy between our findings and these studies may potentially be explained by differences in sampling protocols. The studies conducted by other groups collected plasma for analysis of corticosterone levels only at a single time point in the day, and thus, cannot account for total production levels across a circadian cycle as total daily fecal corticosterone levels do. Although data from these previous studies led authors to conclude that maternal corticosterone levels could not be linked to observations of long-term alterations of offspring circadian behavior, SCN clock and

neuropeptide gene expression, nor alterations in offspring HPA axis when raised in costant light conditions the first three weeks post-natal [42–46], our data may indicate otherwise, and thus, further studies in the area are warranted. Additionally, corticosterone also plays a central role as a metabolic hormone, functioning to mobilize substrates for synthesizing carbohydrates and lipids, supporting the demands of lactation [47]. The elevated levels of corticosterone in L100 may reflect greater eating activity in this group, or even potentially greater suckling activity, as it is released as part of the neuroendocrine response to neonate suckling [48].

Diet, light treatments and day of lactation significantly impacted 24 and 12 h rhythms of activity outside of the nest. Data collection began at the start of the dark phase after delivery of the litter and continued over the next 48 h. Mice spent more time in the nest the first 24 h of data collection versus the second 24 h period. Lower activity outside of the nest immediately after parturition likely reflects the establishment of maternal activities related to care and suckling of pups. Mice in the LD group spent less time in their nests at night than during the day, whereas L100 dams stayed in their nests more at night compared to those in the LD group. Dams on HF diets also spent more time inside the nest than CON. These alterations in behavior likely reflect the response to ambient light, and its variation, as well as metabolic-eating drive to support the energetic demands of lactation.

Under normal 24 h cycles of light and dark, mice are biologically programmed to be more active during the dark (nocturnal) phase, as dictated by their circadian rhythm [49]. Time outside the nest reflects activities other than maternal care and rest [50]. Data on feed intake patterns indicated that LD mice ate more during the dark phase, so greater feed intake during this phase of the LD treated mice reflects some of the eating activity. Similarly, the greater amount of time spent out of the nest of the animals on the CON diet reflects the need to eat more grams of food throughout the day, versus the kcal rich HF diet. This interpretation is consistent with the greater intake of grams of CON diet versus HF diet. The L100 group showed less variation in levels of activity outside of the nest between daytime and nighttime hours, and again this was likely partly driven by eating behavior of the dam.

In both the LD and L100 groups, significant 12-hour rhythms of time in the nest and out of the nest were observed. Studies of maternal behavior on the second day of lactation found dams spent approximately 40% of the time exhibiting crouching behavior, indicative of suckling activity in mice [51]. Periodogram analysis of crouching behavior of dams found significant 24-h and 12-h rhythmicity in wild-type mothers. Therefore the rhythms of out of the nest behavior we observed in our study may be driven partly by the neonate metabolic demands.

The 12 h rhythms in the L100 groups may also reflect the phenomenon of split circadian rhythms, where animals divide a single circadian cycle into two 12-hour phases under constant light conditions [52]. This behavior has been documented in various species, including hamsters, rats, and tree shrews. The phases of these split rhythms are approximately 180° apart and are influenced by light intensity [52]. The underlying mechanism is believed to involve the decoupling of circadian oscillators, potentially located outside the suprachiasmatic nucleus (SCN), which reorganize into two distinct components that respond to changes in light [53]. Models suggest that alterations in the period or coupling strength of these oscillators may contribute to the stable emergence of these 12-hour rhythms [54].

Although we found no difference in the percent of time in the nest between L100 and LD, others reported that dams exposed to constant light in the postnatal period spent more time in the nest than animals exposed to constant darkness [55]. Analysis of maternal care quality found that there was no difference between dams exposed to constant light, LD, or constant darkness in nest building, suckling or autogroom behaviors [55], indicating that staying in the nest may be a means of avoiding light. In addition, a study conducted by another group assessed the effect of HF diet on prepregnancy and postpartum maternal behavior. This group

found divergent results in two different studies. In their first study HF diet impacted nest building prepartum, but had no effect in the second [56,57]. Pup retrieval behavior revealed no significant differences in retrieval latency between dams on control and high fat diets on d3 and d4. However, by d5, high-fat-induced obesity significantly impaired pup retrieval in HF mice compared to control mice [56,57].

Dams in HF diet were less active overall compared to the CON group. This distinct behavior likely reflects a lower need to spend time eating to support energetic demands of lactation due to the high caloric density of the HF diet. Additionally, high-fat diets can negatively impact brain chemistry and reduce activity in mice through inflammation in the hypothalamus and increase oxidative stress [58,59], disruption of gut microbiome which can affect the expression of genes involved in serotonin production and signaling in the brain [60,61], or damaging blood vessels that impair cognitive processes [62,63]. The HF diet has the potential to rapidly change the lipid mediators and gut microbiome composition [64], influencing digestion and fecal production [17,64], that could lead, in our present outcomes, to increase feces weight and corticosterone output levels.

One of the key weaknesses of this study is the lack of behavioral recordings for the L5 group, which limits our ability to fully understand the impact of continuous dim light exposure on maternal behavior and eating patterns. Moreover, due to the use of nesting material, analysis of behavior was limited to time spent in and out of the nest, and did not evaluate the impact of diet and treatments on suckling behavior and other maternal behaviors that potentially could be related to differences in litter growth and long-term behavior that have been reported by others. Consequently, this gap in our data restricts a comprehensive understanding of how different light treatments modulate maternal behavior and their downstream effects on health and development. Additionally, the study found no significant difference between day and night corticosterone concentrations, despite variations in fecal output, eating behavior, and time spent inside and outside the nest. This absence of corticosterone variation may be due to differences in sampling protocols between experiments, variations in physiological states, or both, potentially obscuring diurnal fluctuations in corticosterone levels and their relationship with observed behavioral changes. Future studies should include detailed behavioral analyses for all treatment groups and employ more frequent sampling intervals to better elucidate the complex interactions between light exposure, diet, maternal behavior, and the circadian dynamics of corticosterone secretion.

## Conclusion

During pregnancy and lactation, maternal circadian rhythms are maintained, but relative to non-reproductive states, 24 h rhythms of hormones, gene expression and behavior are attenuated. Daily rhythms become modified in response to offspring metabolic cues and maternal care behaviors, ensuring the survival of neonates. Alterations in environment and diet alter maternal behavior and physiology, and if profound enough, negatively impact offspring development and their long-term health. Humans are similarly sensitive to light, with light exposure during the night producing the most significant phase shifts. This sensitivity extends to both intensity and timing of light, where varying light intensities and specific phases of exposure result in different magnitudes of phase shifts. Thus, light has a critical role in maintaining circadian synchrony. Exposure to constant light or a HF diet disrupts circadian rhythms, leading to desynchronization between the internal clock and environmental cues, causing alterations in metabolic cycles and energy homeostasis, which in turn affect the health of the individual and the development of their progeny. Data collected in this study demonstrate the intricate interplay between diet, light exposure, and physiological state in shaping maternal behavior

and metabolic rhythms during pregnancy and lactation in mice. Our findings revealed that a HF diet and continuous light exposure significantly alter feeding patterns, fecal output, corticosterone levels, and activity behaviors of dams. The increased caloric intake and altered feeding times associated with the HF diet, along with continuous light exposure, underscore the profound impact of environmental and dietary factors on circadian rhythms and metabolic processes of pregnant and lactating female mice. The disruption of typical light-dark cycles and the consumption of a HF diet led to modifications in feeding behavior, fecal corticosterone output, and time spent in and out of the nest. These changes in maternal behavior, hormones and metabolism impact the developmental environment of their offspring, spanning from their care, nutrition, and hormone exposures, and likely underlie some of the long-term programming effects of maternal high-fat diet and constant light exposures on rodent offspring. The findings may provide potential understanding of results from human studies, as elevated cortisol in humans is linked to obesity and shift-work stress and, adverse outcomes of maternal and offspring health.

## Supporting information

**S1 File. Post-hoc analysis for interaction of diet, light, period of the day, and period of physiological state.**
(XLSX)

**S2 File. Supplementary information–dataset.**
(XLSX)

## Author Contributions

**Conceptualization:** Theresa Casey.

**Data curation:** Leriana Garcia Reis, Kelsey Teeple, Jenna Lynn Schoonmaker, Corrin Davis, Sara Scinto.

**Formal analysis:** Leriana Garcia Reis, Corrin Davis, Sara Scinto, Allan Schinckel.

**Investigation:** Corrin Davis.

**Methodology:** Jenna Lynn Schoonmaker.

**Project administration:** Kelsey Teeple, Theresa Casey.

**Supervision:** Kelsey Teeple, Theresa Casey.

**Validation:** Theresa Casey.

**Visualization:** Allan Schinckel, Theresa Casey.

**Writing – original draft:** Leriana Garcia Reis.

**Writing – review & editing:** Corrin Davis, Sara Scinto, Theresa Casey.

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
