## [Decision Letter · Decision Letter 0]

26 Aug 2024

PONE-D-24-29736Constant light and high fat diet alter daily patterns of activity, feed intake and fecal corticosterone levels in pregnant and lactating female ICR micePLOS ONE

Dear Dr. Casey,

Thank you for submitting your manuscript to PLOS ONE. After careful consideration, we feel that it has merit but does not fully meet PLOS ONE’s publication criteria as it currently stands. Therefore, we invite you to submit a revised version of the manuscript that addresses the points raised during the review process.

**ACADEMIC EDITOR: **

-please do follow very useful comments of reviewer's to improve the manuscript 

We look forward to receiving your revised manuscript.

Kind regards,

Prof. Dr. Dragan Hrncic, MD, PhD 

Academic Editor

PLOS ONE

Reviewers' comments:

Reviewer's Responses to Questions

**Comments to the Author**

1. Is the manuscript technically sound, and do the data support the conclusions?

Reviewer #1: Yes

2. Has the statistical analysis been performed appropriately and rigorously? 

Reviewer #1: Yes

3. Have the authors made all data underlying the findings in their manuscript fully available?

Reviewer #1: Yes

4. Is the manuscript presented in an intelligible fashion and written in standard English?

Reviewer #1: Yes

5. Review Comments to the Author

Reviewer #1: Overall this is a very well written and interesting study. It is well described and well designed.

There are some concerns should be addressed in the discussion:

1) While there were no changes observed in cortisol, did you explore any of the other stress hormones/pathways? (ie, epinephrine, norepinephrine?), or photoperiod hormones (melatonin, prolactin?).

2) There is no discussion about the phentoypes that can be observed during lactation where there is no actually adiposity ("lean" so to speak) in animals during lactation due to the energy associated with lactation, while sometimes there is. Where there any differences in adiposity between diets, diets*light exposure, or light exposure? How might this be related to outcomes of your study if at all?

3) Did you assess any maternal type behaviors? like pup retrieval and return to the nest? Were these affected by the treatments?

These would immensely help the discussion and to support/refute findings in the data. The work is really good and this will advance the field.

6. PLOS authors have the option to publish the peer review history of their article (what does this mean?). If published, this will include your full peer review and any attached files.

Reviewer #1: No

---

## [Author Response · Author response to Decision Letter 0]

31 Aug 2024

Response to Manuscript Reviewers: “Constant light and high fat diet alter daily patterns of activity, feed intake and fecal corticosterone levels in pregnant and lactating female ICR mice”

We would like to thank you for taking the time to review our manuscript, your comments served to be insightful and addressing them increased its quality.

The authors confirmed that the submission contains all raw data required to replicate the results of their study – S2 file.

Reviewer #1:

1) While there were no changes observed in cortisol, did you explore any of the other stress hormones/pathways? (ie, epinephrine, norepinephrine?), or photoperiod hormones (melatonin, prolactin?).

Response: Fecal corticosterone was measured as a biomarker of stress due to its well-established role in reflecting the hypothalamic-pituitary-adrenal (HPA) axis activity in rodents. We previously reported findings regarding serum prolactin (Teeple et al., 2023, doi:10.1242/BIO.060088/333487), which was significantly elevated in the LL treatment (see lines 65-69 in the Introduction). Melatonin was not measured in our study as a marker of photoperiod response, as ICR mice are homozygous for B6J Hiomt mutant alleles, rendering them unable to produce melatonin and thus melatonin deficient (Kasahara, T., et al 2010, doi.org/10.1073/pnas.0914399107). 

2) There is no discussion about the phentoypes that can be observed during lactation where there is no actually adiposity ("lean" so to speak) in animals during lactation due to the energy associated with lactation, while sometimes there is. Where there any differences in adiposity between diets, diets*light exposure, or light exposure? How might this be related to outcomes of your study if at all?

Response: We have revised the manuscript (Lines 315-323) to better discuss these outcomes. In the present study, we focused on the effects of a high-fat diet and continuous light exposure on daily eating patterns, maternal activity, and corticosterone output. We previously addressed adiposity-related aspects in our prior studies (Teeple et al., 2023, doi:10.1242/BIO.060088/333487; Teeple et al., 2023, doi:10.1371/JOURNAL.PONE.0279209), finding that HF diet mice had higher BMI by lactation day 12, but adiposity differences diminished as they mobilized energy stores during lactation. Our findings align with the concept that HF dams, despite having higher BMI and percent body fat prior to lactation, mobilized their extra energy stores during lactation, leading to diminished differences in adiposity by the study's end. In contrast, CON mice likely had to rely on increased food consumption to meet the energy demands of lactation, which aligns with the observation that pre-pregnancy obesity in human models does not typically result in significant changes in BMI during lactation (Marshall et al., 2022 doi: 10.1016/j.ajog.2021.12.035; Winkvist and Rasmussen, 1999 doi: 10.1023/A:1018706131168).

3) Did you assess any maternal type behaviors? like pup retrieval and return to the nest? Were these affected by the treatments?

Response: We agree that assessment of maternal behaviors like pup retrieval and nest building would have strengthened the study, however, we decided to leave nesting material in the cage and limited behavior assessment to activity patterns in a relative non-invasive manner. We discuss this as a limitation of our study (lines 402-410, 420-434).

---

## [Decision Letter · Decision Letter 1]

7 Oct 2024

Constant light and high fat diet alter daily patterns of activity, feed intake and fecal corticosterone levels in pregnant and lactating female ICR mice

PONE-D-24-29736R1

Dear Dr. Casey,

We’re pleased to inform you that your manuscript has been judged scientifically suitable for publication and will be formally accepted for publication once it meets all outstanding technical requirements.

Kind regards,

Prof. Dr. Dragan Hrncic, MD, PhD

Academic Editor

PLOS ONE

Additional Editor Comments (optional):

Reviewers' comments:

Reviewer's Responses to Questions

**Comments to the Author**

1. If the authors have adequately addressed your comments raised in a previous round of review and you feel that this manuscript is now acceptable for publication, you may indicate that here to bypass the “Comments to the Author” section, enter your conflict of interest statement in the “Confidential to Editor” section, and submit your "Accept" recommendation.

Reviewer #1: All comments have been addressed

2. Is the manuscript technically sound, and do the data support the conclusions?

Reviewer #1: Yes

3. Has the statistical analysis been performed appropriately and rigorously? 

Reviewer #1: Yes

4. Have the authors made all data underlying the findings in their manuscript fully available?

Reviewer #1: Yes

5. Is the manuscript presented in an intelligible fashion and written in standard English?

Reviewer #1: Yes

6. Review Comments to the Author

Reviewer #1: Thank you for addressing my comments and updating the manuscript for clarification. You have addressed all the concerns I raised.

7. PLOS authors have the option to publish the peer review history of their article (what does this mean?). If published, this will include your full peer review and any attached files.

Reviewer #1: No

---

## [Editor Report · Acceptance letter]

14 Oct 2024

PONE-D-24-29736R1 

PLOS ONE

Dear Dr. Casey, 

I'm pleased to inform you that your manuscript has been deemed suitable for publication in PLOS ONE. Congratulations! Your manuscript is now being handed over to our production team.

Kind regards, 

on behalf of

Professor Dragan Hrncic 

Academic Editor

PLOS ONE